# Decoupling Generalizability and Membership Privacy Risks in Neural Networks

## Abstract

A deep learning model usually has to sacrifice some utilities when it acquires some other abilities or characteristics. Privacy preservation has such trade-off relationships with utilities. The loss disparity between various defense approaches implies the potential to decouple generalizability and privacy risks to maximize privacy gain. In this paper, we identify that the model's generalization and privacy risks exist in different regions in deep neural network architectures. Based on the observations that we investigate, we propose *Privacy-Preserving Training Principle* (PPTP) to protect model components from privacy risks while minimizing the loss in generalizability. Through extensive evaluations, our approach shows significantly better maintenance in model generalizability while enhancing privacy preservation.

## 1 Introduction

A machine learning model acquires accurate recognition abilities by learning how to fit the training data points. However, even if a model shows a good performance for an objective of a given task, it may suffer from the risk of data privacy leakage, especially in privacy-sensitive applications or systems. The concern about this issue is also raised by several existing studies: Arpit et al. (2017); Chatterjee (2018) showed neural networks' potential privacy risks via fitting on random data. Shokri et al. (2017) showed that privacy risks are widely posed in neural networks by showing the possibility of black-box membership inference attacks (MIAs). Further studies Choquette-Choo et al. (2021); Del Grosso et al. (2022) showed that a model has significant behavioral differences between the training and testing data in various aspects, especially in robustness. Also, Stephenson et al. (2021) found that the model's memory of training data points becomes solid as training progresses. All of these prior observations and knowledge indicate that machine learning models have a strong memory for data, which enables the model to achieve near-perfect performance on the training dataset.

Since a well-trained model will retain a lot of data traces, deploying a model in a privacy-sensitive system requires particular caution. This risk occurs not only in classification models Shokri et al. (2017); Song & Mittal (2021); Choquette-Choo et al. (2021) but also in some other machine learning domains, such as generative models Chen et al. (2020) or transfer learning Zou et al. (2020); Wu et al. (2024), etc. Besides, models could leak privacy in various ways, e.g., membership inference attacks Shokri et al. (2017), model inversion attacks Fredrikson et al. (2015), and model extraction attacks Tramèr et al. (2016). This universality makes the output of any model pose the potential risk of privacy leakage. Therefore, understanding the sources of privacy risks and how to rectify them is the key to strengthening the model to be trustworthy.

In this paper, we study where and how layer-level privacy leakage occurs in neural network architectures. We empirically identify that the model's generalizability and privacy risk are separable. Then, we propose Privacy-Preserving Training Principle (PPTP) to minimize negative impacts on the model utility during training with privacy defense approaches. Here is a brief overview of our novel observations and contributions:

- We structurally and precisely investigate where and how the machine learning model produces privacy risks, and identify that ***privacy risks and generalizability occur in different regions in a model.***

- We propose a new cost-effective training paradigm for utility-privacy trade-offs. It ***decouples utility and privacy*** into two separate parts, enabling the model to outperform existing privacy defense approaches.

- Our approach bridges the gap between generalizability and privacy training, empowering the model to choose its proper training approaches (for both utility and privacy) based on its application contexts and helping the model overcome the limitations of the existing privacy defense approaches.

## 2  Related Work

There have been various studies to prevent the ML model from privacy risks. DP-SGD Abadi et al. (2016) tried to train the model with a less fitting degree via additional noises in the optimizer. Nasr et al. (2018) developed an adversarial mechanism to help the model obtain aligned predictions. However, both require significantly increasing training costs due to their limitations in parallel computations. Jia et al. (2019); Yang et al. (2023) proposed decorators to reproduce better-aligned predictions without retraining the model. Shejwalkar & Houmansadr (2021) tried to mitigate the privacy risk via knowledge distillation. Tang et al. (2022) further improved the utilization rate of training data via ensemble-based knowledge distillation. Li et al. (2021) attempted the alignment between training and non-training accuracy during training. Chen et al. (2022) proposed an efficient training paradigm with effective privacy-preserving ability. Fang & Kim (2024a;b) discussed the privacy issues in the bottleneck layer. Liu et al. (2024) observed the convexity of loss functions is a factor of privacy leakage and tried to mitigate it with a concave term. Although the conclusions are not entirely consistent, Wang et al. (2021); Yuan & Zhang (2022) explored the impact of network pruning on privacy risks. Kaya & Dumitras (2021); Yu et al. (2021) explored which data augmentation techniques are beneficial to privacy. Tan et al. (2023) pointed out that higher model dimensions are possibly more privacy-risky. Li et al. (2024) tried to focus on the most privacy-risky data points. Carlini et al. (2022b) found that a model always experiences varying degrees for different data points in traditional training settings. Zhang et al. (2024) found some components lead to machine learning models at severe privacy risks.

Despite considerable progress, it still remains a work in progress. Like obtaining other characteristics (e.g., Robustness Goodfellow et al. (2015), Fairness Mehrabi et al. (2021), and Data Unlearning Bourtoule et al. (2021)), models always inevitably have to pay the price of utility for privacy with current approaches. Sometimes, this expense is even too high for the model to maintain reasonable performance due to the inherent characteristics of the dataset Carlini et al. (2022a). We question the preconceived view kept by current privacy-preserving approaches that the model is a privacy-risky entity and explore where the model produces the privacy risks in the next section.

## 3  Does Prediction Disparity Exist Everywhere?

### 3.1  Why the Question Matters

Recent non-decorator defense approaches Abadi et al. (2016); Nasr et al. (2018); Shejwalkar & Houmansadr (2021); Chen et al. (2022); Tang et al. (2022) usually train a model from scratch while decorator approaches Jia et al. (2019); Yang et al. (2023) usually add extra filters to the model externally rather than training the model itself. A direct advantage of training from scratch is that it is implementation-friendly. However, training the whole model could lead to an unnecessary utility loss as we see a discrepancy between adversarial training on the whole model Shafahi et al. (2019) and training the last layer only Kirichenko et al. (2023). Then, it gives us a question: **Does prediction disparity exist everywhere**? In this section, we discuss the correlation between privacy-risk and other attributes of machine learning models, such as feature map size, channel size, and depth.

It is known that the model's generalizability is affected by the depth Baldock et al. (2021). That is, a model gradually obtains generalizability layer by layer while learning samples with various difficulty levels. If privacy risk accompanies the learning of different difficulty-level samples, the privacy risk should exist in each layer and "*gradually*" show more and more prediction disparity. Meanwhile, Kirichenko et al. (2023) showed that spurious correlations can be mitigated by retraining only the last layer, hinting that different

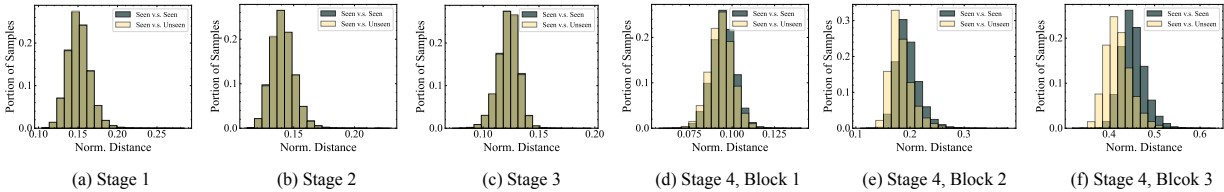

| (a) Stage 1 | (b) Stage 2 | (c) Stage 3 | (d) Stage 4, Block 1 | (e) Stage 4, Block 2 | (f) Stage 4, Blcok 3 |

Figure 1: The sample-level feature map differences (norm. distance on x-axis). No disparity is observed in Stage 1–3, whereas gradually increasing disparity is observed within Stage 4 (ResNet152, TinyImageNet, data augmented)

layers may not exhibit the same characteristics with regard to generalizability. Such recent insights challenge us to locate where a disparity between generalizability and privacy risks starts occurring.

## 3.2 Sample-Level Measurement Design

Unlike the logits produced by the classification layer, there is no direct way to know what a confident feature map should look like. Therefore, we develop a method to examine distribution disparity. Let $D_{all}$ denote the entire training set. We split $D_{all}$ into two halves: $D_{h1}$ and $D_{h2}$. Then, we train two models, $M_{all}$ and $M_{h1}$, with the same architecture and configurations with training sets $D_{all}$ and $D_{h1}$, respectively. Therefore, every data point $x_1 \in D_{h1}$ is seen data for both two models, while $x_2 \in D_{h2}$ is seen data for only $M_{all}$ but unseen data for $M_{h1}$.

Then, we compute the feature map differences using Euclidean distance $\texttt{Dist}(p,q) = \sqrt{\sum_{i=1}^{d}(p_i - q_i)^2/d}$, where $d$ is the number of dimensions of the feature maps and $1/\sqrt{d}$ is an extra normalized item, while $p$ and $q$ denote the feature maps produced by $M_{all}$ and $M_{h1}$ on the same input, respectively. If a layer does not produce prediction disparity, then the feature map distance distributions on $D_{h1}$ and $D_{h2}$ should be equivalent, and vice versa.

## 3.3 Empirical Verification

First of all, we need to check if the disparity happens in all layers or not. Fig. 1 plots sample-level feature map differences of seen and unseen data (norm. distance on x-axis) where we find that the disparity **abruptly** starts in stage 4. In other words, there is only a small portion (the 4th stage contains only 3 blocks while the entire model contains 33 blocks) of ResNet152 that leaks privacy, identifying that the model is not globally privacy-risky, but *regionally*. However, what conditions cause disparity is not clear yet. Hence, we explore different factors' impacts on disparity in the rest of this section. We explore these factors using three different architectures as shown in Fig. 2: VGG Simonyan & Zisserman (2015b), ResNet He et al. (2016), and Active Token Mixer (ATM) (which is a

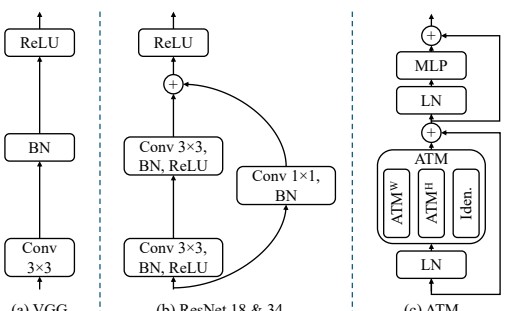

Figure 2: Overview of the three architectures' backbone modules.

transformer-type architecture) Wei et al. (2023). In the three architectures, ResNet and ATM have residual connections, while VGG does not. ATM has self-attention-type computation modules, which differentiates it from VGG and ResNet.

**Data Augmentation** Kaya & Dumitras (2021); Yu et al. (2021) found some data augmentation techniques are beneficial to the model's privacy. Figure 3 shows that data augmentation does help mitigate disparity when it occurs (please compare (d) and (h)). However, whether with or without data augmentation, disparity happens in stage 4, which exhibits that data augmentation has no impact on "where" disparity happens.

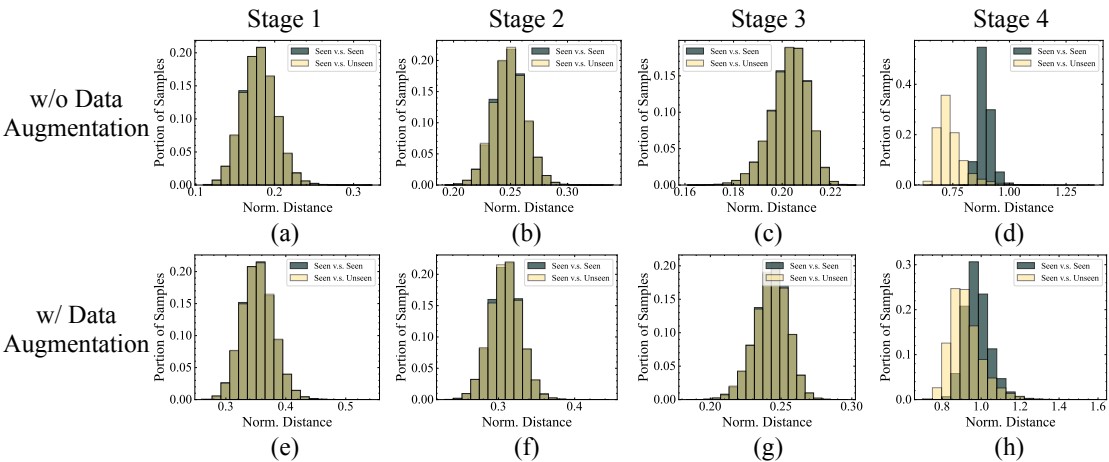

Figure 3: Comparison of models trained with and without data augmentation. (ResNet18, TinyImageNet)

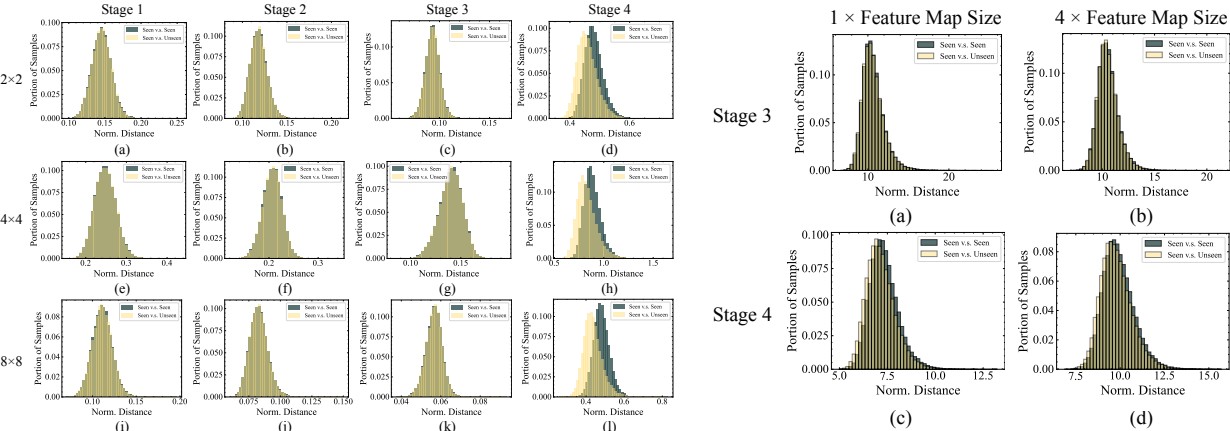

Figure 4: Comparison of ResNet18 with different feature map sizes in the 4th stage. (TinyImageNet, data augmented)

Figure 5: Comparison of ATM-XT in various channel sizes at the 3rd & 4th stage. (TinyImageNet, data augmented).

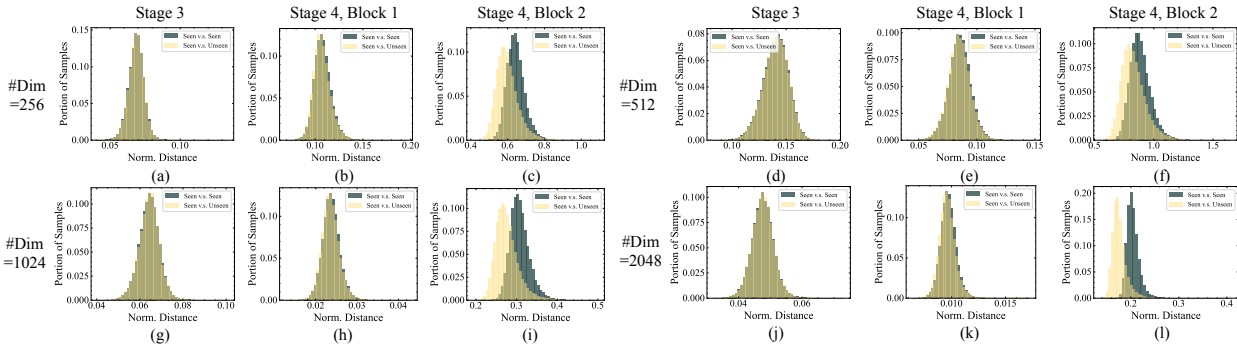

Figure 6: Comparison of ResNet18 in various channel sizes at the 4th stage. (TinyImageNet, data augmented)

**Feature Map Size**   One significant change along with stages is the feature map size. After each downsampling layer at the head of each stage in the ResNet (similar to many widely used architectures such as CNN, ViT, Mixer, etc.), the feature map's width and height are halved. To study how generalizability

Table 1: The performance comparison among different feature map sizes at the 4th stage. (ResNet18, TinyImageNet, data augmented)

| Accuracy (%) | Feature map size | | |
|---|---|---|---|
| | $2 \times 2$ | $4 \times 4$ | $8 \times 8$ |
| Train | 99.98 | 99.98 | 99.98 |
| Test | 52.32 | 54.58 | 61.53 |

Table 2: The performance comparison among different feature map sizes at the 4th stage. (ATM-XT, TinyImageNet, Data Augmentation)

| Accuracy (%) | $1\times$ Feature Map Size | $4\times$ Feature Map Size |
|---|---|---|
| Train | 99.97 | 99.98 |
| Test | 36.28 | 47.29 |

Table 3: The testing accuracy (%) comparison among different training data and different channel sizes at the 4th stage. (ResNet18, TinyImageNet, data augmented)

| Accuracy (%) | Channel size | | | |
|---|---|---|---|---|
| | 256 | 512 | 1024 | 2048 |
| Test | 52.52 | 54.58 | 53.99 | 55.03 |

and disparity change along with feature map size, the feature map size at the 4th stage of ResNet18 is enlarged and reduced (originally $4 \times 4$). By considering both Figure 4 and Table 1 together, we find that, while enlarging the feature map also has no impact on where the disparity happens, it enhances the model's generalizability while not significantly exacerbating the disparity. The same trends are also observed in ATMs (refer to Figure 5 and Table 2)

**Channel Size** One common design trend in deep learning in the recent decade is increasing channel size. As shown in Fig. 6 and Table 3, the disparity becomes more and more significant as channel size increases while changes in testing accuracy are not as significant as the changes by enlarged feature map size (refer to Table 1) at 4th stage. That is, too many channel sizes make the model prone to produce disparity, leading to more severe privacy risks. Therefore, it is not worth designing too many channels from the point of view of utility-privacy trade-offs.

**Depth** As observed above, since feature map size and channel size do not impact where privacy risk occurs, now we investigate the depth of the model. First, we use a support vector machine (SVM) Cortes & Vapnik (1995) to classify the features extracted in each stage (see Figure 7). We notice that some stages have limited (or even no) benefits to the model's generalizability. The testing accuracy in all models keeps consistently increas-

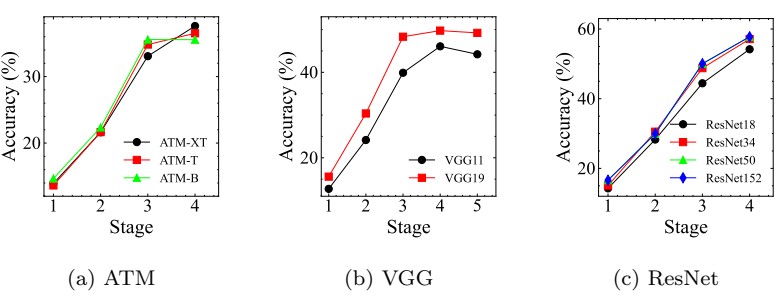

| (a) ATM | (b) VGG | (c) ResNet |
|---|---|---|

Figure 7: Testing accuracy changes along with stages using SVM. (TinyImageNet)

ing in the early stages (stages 1, 2, and 3 in Figure 7). However, this growth rate starts to level off after stage 3 to various degrees - some of them slow down (ResNet), and others stagnate or even deteriorate (ATM and VGG). Please note that this stagnation of growth perfectly matches where the privacy risk occurs (e.g., Figure 8 & Figure 7(b)). It explains that when the model cannot learn sufficient features to be generalized enough as much as its capacity, it instead learns many ineffective features. These features cannot help generalizability but can help the model fit better on the train set. Therefore, these features must be privacy-risky since they are only valid on the train set.

Besides, another notable point is that VGG shows different trends (privacy risks occur at different layers in VGG11&19, see Figure 8) from ATM's case (privacy risks always start at the 3rd stage, see Figure 10) and ResNet's case (privacy risks always start at the 4th stage, see Figure 9). That is because VGG11 can maintain the growing speed of generalizability between stages 3 and 4 while VGG19 cannot. In contrast, ResNet never shows privacy risks before the 4th stage even if most additional layers from ResNet18 to ResNet152 belong to the 3rd stage. This phenomenon further explains the relationship between generalized features and privacy.

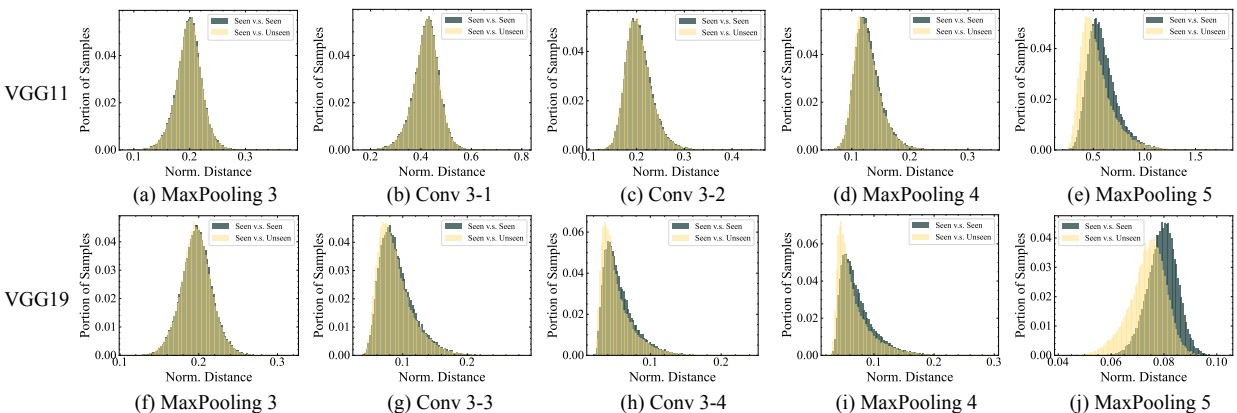

Figure 8: The sample-level feature map differences measurement. The disparity is observed earlier in VGG19. (VGG11&19, TinyImageNet, data augmented)

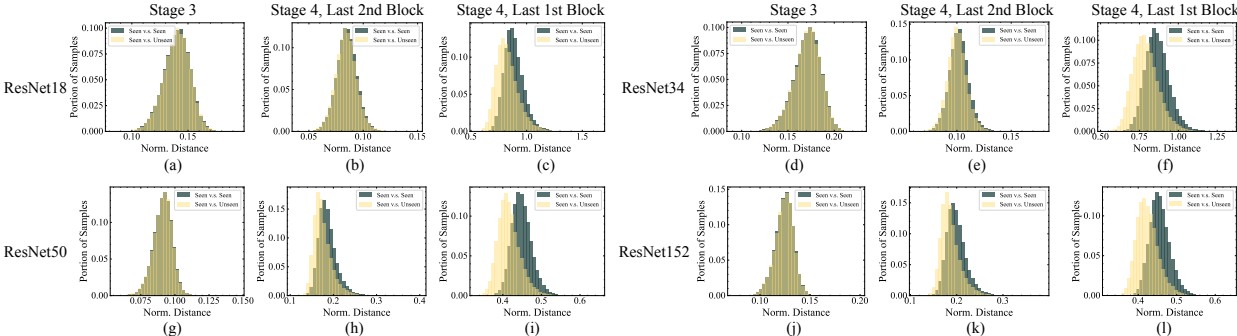

Figure 9: Comparison of feature map differences among different ResNet depths.

**Does Prediction Disparity Exist Everywhere?** The answer is no! In short, privacy risks only exist *in certain components without significant contributions to the model's generalizability.* Summarizing the observations above, we can draw the following insights of privacy and generalizability of an architecture:

- **Privacy risk and generalizability are separable**: in standard training procedures, privacy-risky features are learned together with generalized features due to imperfect or inadequate designs in various aspects, such as loss function, label, and model architecture). Fortunately, they exist in different regions of neural networks, identifying the feasibility of decoupling them.

- **Non- or less-generalized features lead to privacy risks**: Less-generalized features are usually learned in later layers. Although the increase in model computation capacity (i.e., increase in depth, channel size, and feature map size) brings the model effective improvements in generalizability, the surplus non- or less-generalized features put the model at higher privacy risks.

- **Privacy risk occurs in later stages**: The less-generalized features mainly exist in later stages, which is also consistent with the observation of Baldock et al. (2021). The more challenging to learn the feature is, the more possibly under-generalized the feature is. That is, privacy risk mostly exists in the later stage of the neural network.

These insights identify an issue in the current privacy defense approaches: they treat the model as a whole during training without distinguishing the privacy risks of various components. With no doubt, disturbing generalized privacy-safe features will lead to the unnecessary deterioration of the model's generalizability. Hence, it is necessary to develop a more considerate training paradigm with regard to privacy.

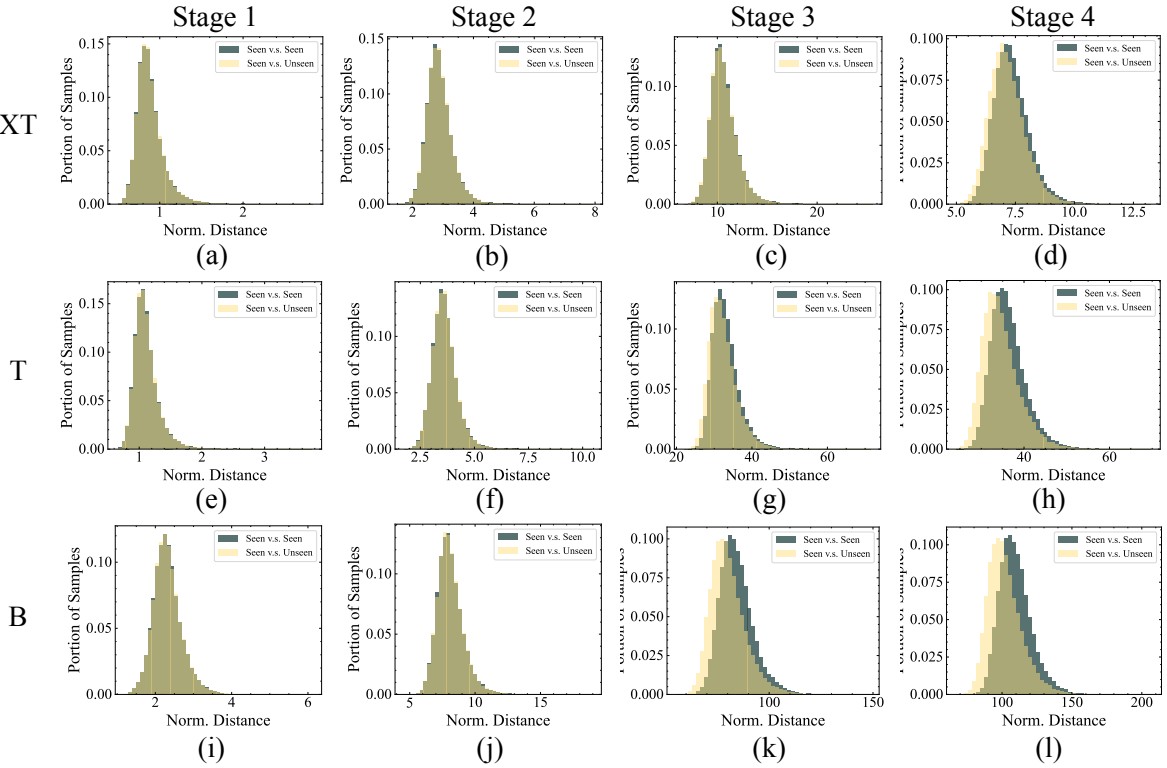

Figure 10: Comparison of models in various depths. XT, T, and B denote ATM-XT, -T, and -B, respectively. (ATM, TinyImageNet)

## 4 Privacy-Preserving Training Principle

With the observations above in mind, we obtain the insight that a model does not need to update all weights to achieve better privacy-utility trade-offs. Instead, only privacy-risky layers can be retrained in a privacy-safe way. To achieve this, we propose a retraining approach, *Privacy-Preserving Training Principle* (PPTP), to enable a model to minimize utility loss while obtaining privacy. The approach is demonstrated in Alg. 1. In the algorithm, we first determine the privacy-risk and -safe layers. Then, *the weights of privacy-safe layers are frozen since they have learned generalized and privacy-safe features.* After that, the weights of the privacy-risky layers are rewound for retraining. Because general training approaches do not take into account privacy criteria, these layers need to be trained with a privacy-defending approach to be privacy-safe. With our approach, PPTP, the model can maintain highly generalized features while reducing privacy-risky features. We empirically show its effectiveness by comparing differences between training with and without our approach in the next section.

---

**Algorithm 1** Privacy-Preserving Training Principle (PPTP)

---

**Input**: Training Dataset $\mathcal{D} = \{(x_i, y_i)\}_{i=1}^N$, Ordinarily Pre-Trained and Privacy-Risky Model $M_{pr}$ and its Parameters $\theta$, Privacy-Preserving Training Approach $f$, Training Epochs $E$, and Other Retraining Configurations $C$;

**Output**: Privacy-Safe Model $M_{ps}$;

1: Split the model with parameters $\theta$ into Privacy-Safe and -Risk Layers, denoting their parameters as $\theta_{ps}$ and $\theta_{pr}$, respectively.
2: Freeze Privacy-Safe Parameters $\theta_{ps}$
3: Rewind Privacy-Risky Parameters $\theta_{pr}$
4: **for** *epoch* **in** $\{1, 2, \cdots, E\}$ **do**
5:     Retrain the model with privacy-preserving training approach $f(\mathcal{D}, M_{pr}, C)$
6: **end for**
7: Return privacy-safe model $M_{ps}$

---

Table 4: Hyper-parameters searching space.

| Hyper-Parameter | Adv-Reg | SELENA | RelaxLoss |
|---|---|---|---|
| $\alpha$ | $[1.0, 5.0]$ | N/A | $[1.0, 3.0]$ |
| $k$ | 3 | 25 | N/A |
| $l$ | N/A | $[3, 10]$ | N/A |

Table 5: The information about computation environment.

| | OS | CPU | RAM | GPU | CUDA | Pytroch |
|---|---|---|---|---|---|---|
| Information | Ubuntu 22.04 | 96 cores | 460 GB | V100 | 12.1 | 2.1 |

## 5 Experiments

### 5.1 Experimental Settings

We evaluate our approach and others on CIFAR-100 Krizhevsky et al. (2009) and TinyImageNet Le & Yang (2015). We evaluate our approaches on these two datasets to show the effectiveness on small and large datasets since extensive studies Chen et al. (2018); Yun et al. (2020) have demonstrated that methods that work well on small datasets may have insignificant improvement on larger datasets. Besides, the layers to be frozen in a model may vary over different datasets, especially with different input dimensions or information domains. In other words, the model may produce privacy risks earlier or later depending on the dataset. Data augmentation techniques, including random flipping & cropping Simonyan & Zisserman (2015a), are applied when training the model with a privacy-defending technique. As for privacy attacks, we evaluate our approach and others on correctness-based MIAs Yeom et al. (2020), confidence-based MIAs Yeom et al. (2018); Song et al. (2019); Song & Mittal (2021), entropy-based MIAs Shokri et al. (2017); Song & Mittal (2021), modified-entropy-based MIAs Song & Mittal (2021), and neural network based MIAs Shokri et al. (2017); Nasr et al. (2018). We include privacy training techniques such as adversarial regularization (`Adv-Reg`) Nasr et al. (2018), SELENA Tang et al. (2022), and relaxed loss (`RelaxLoss`) Chen et al. (2022). For each result, we execute three independent runs to ensure its stability. The hyper-parameters searching space of cited privacy defense approaches are shown in Table 4. Also, the main information of experimental software and hardware environment is presented in Table 7.

### 5.2 Results and Discussions

#### 5.2.1 Ablation Study

To change the characteristics of a trained model, we can consider the two options, weight rewinding and fine-tuning. As seen in Algorithm 1, we choose rewinding and retraining to achieve more advanced utility-privacy trade-offs. Most privacy defense approaches, such as Nasr et al. (2018); Shejwalkar & Houmansadr (2021); Chen et al. (2022), are designed for training from scratch. Thus rewinding is more adequate for them. Besides, it is necessary to check if retraining privacy-risky

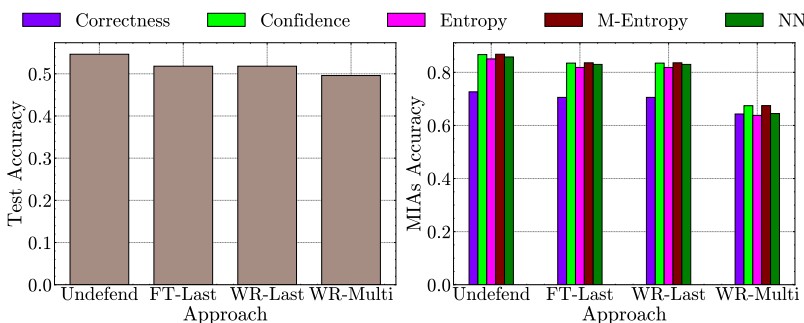

Figure 11: Comparison of retraining the model with different designs (TinyImageNet, ResNet18).

layers can help the model achieve better utility-privacy trade-offs. To examine it, we show the performance comparison between training from scratch, retraining privacy-risky layers, and retraining the classification layers using RelaxLoss. As seen in Figure 11, retraining with the last layer only, regardless of whether through fine-tuning or weights rewinding, cannot effectively mitigate the model's privacy risks, while retraining multiple privacy-risky layers helps the model mitigate the privacy risks successfully.

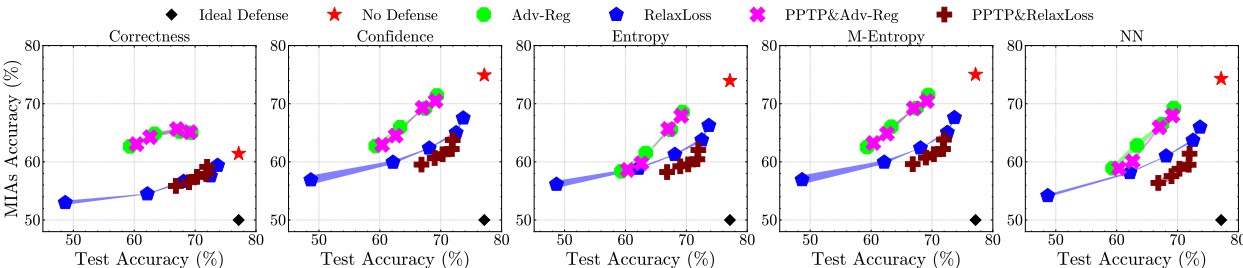

Figure 12: Comparisons with existing privacy-preserving techniques (CIFAR100, ResNet18).

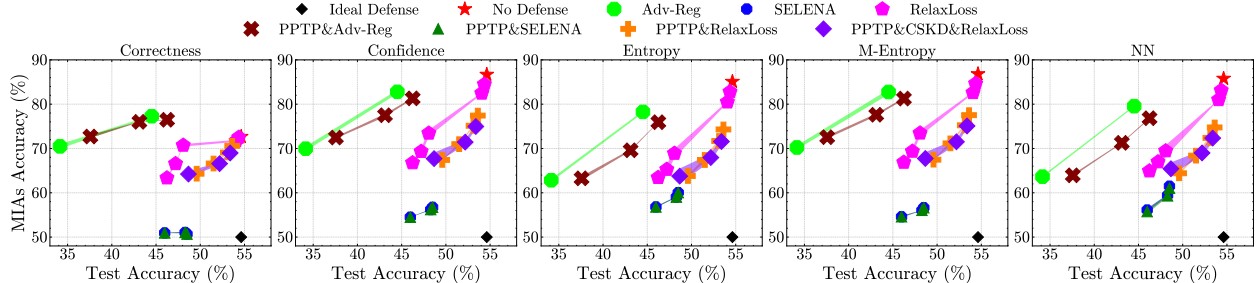

Figure 13: Comparisons with existing privacy-preserving techniques (TinyImageNet, ResNet18).

### 5.2.2 Comparisons with Other Approaches

In CIFAR-100, we evaluate our approach with Adv-Reg and RelaxLoss. As shown in Figure 12, both RelaxLoss and Adv-Reg show improvement when they are used to retrain the model with PPTP. However, the performance of correctness-based MIAs does not show significant improvement. PPTP retains the partially generalized features learned by CE to help the model maintain its utility, whereas it does not benefit the correctness alignment on train and test sets. Fortunately, however, this does not mean that PPTP cannot work with MIAs based on robustness or fairness. For instance, robustness issues of neural networks mainly exist in the final layer Kirichenko et al. (2023), meaning that the adversarial disparity can be also mitigated in the later layers to defend the robustness-based MIAs such as Del Grosso et al. (2022). We do not report the results of SELENA on this dataset since gradient explosions frequently occur on SELENA in the distillation stage (this phenomenon also occurs on TinyImageNet, although less frequently.)

In TinyImageNet, we evaluate our approach with Adv-Reg, SELENA, and RelaxLoss. Compared with results on CIFAR-100, the trends on TinyImageNet (Fig. 13) vary a lot. The Adv-Reg and RelaxLoss show more significant improvement once PPTP is applied. The results identify that our approach is effective in maintaining good performance on larger datasets. An important factor

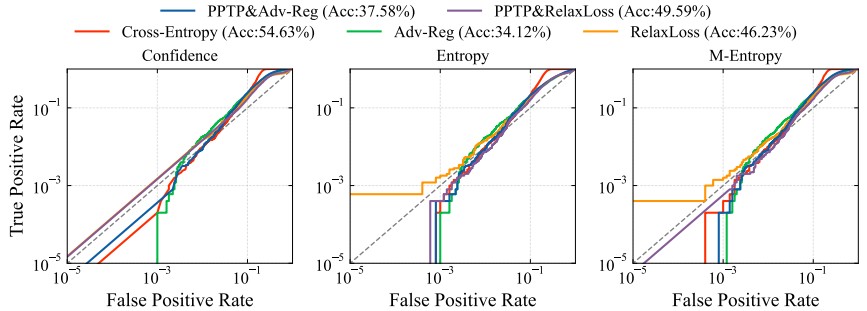

Figure 14: AUC-ROC curve comparison of defense approaches with and without PPTP under various MIAs.

is that we freeze much more weights in TinyImageNet than in CIFAR-100 because the disparity occurs in different layers (see Table 6). This also shows the potential of the model compression methods Zhang et al. (2019; 2022), which can help the model achieve better utility performance at earlier stages, to be applied for privacy preservation. In contrast, the improvement in SELENA is very slight. We factor in

Table 6: Weight-freezing stopping layers

| Model | CIFAR-100 | TinyImageNet |
|---|---|---|
| ResNet18 | conv4_x (stage 3) | conv5_x (stage 4) |

Table 7: The performance comparison of different training data and training techniques. (ResNet18, TinyImageNet, data augmented)

| Accuracy (%) | CE & Full Data | CE & Half Data | RxL & Full Data |
|---|---|---|---|
| Train | 99.98 | 99.99 | 79.68 |
| Test | 54.58 | 44.07 | 47.57 |

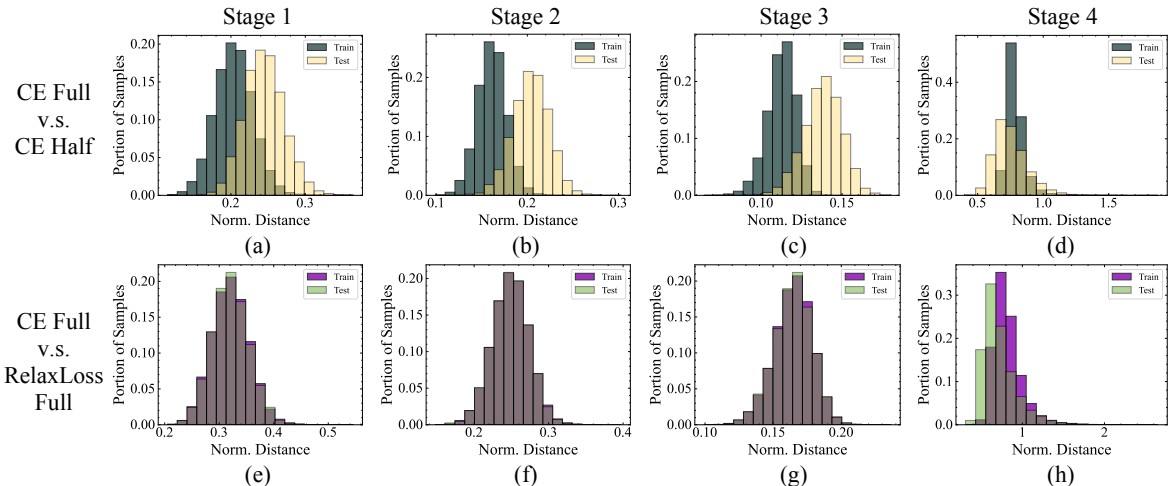

Figure 16: The distribution differences between training full data with CE, half data with CE, and full data with RelaxLoss.

that data augmentation is disabled in the distillation phase - if data augmentation was applied in this phase, the training time cost could have been over ten times greater than the current one. Additionally, it exhibits that our approach decouples the utility and privacy training. To demonstrate this characteristic, we apply class-wise self-knowledge distillation (`CSKD`) Yun et al. (2020) to pre-train the model. Compared with privacy training from scratch, pre-training with `CSKD` also shows improvements in utility at the same privacy level. Finally, to further exhibit privacy improvement using our approach, we evaluate Adv-Reg and RelaxLoss with and without PPTP and plot the AUC-ROC curve Carlini et al. (2022a) for TinyImageNet. As shown in Figure 14, compared with original Adv-Reg and RelaxLoss, our approach achieves comparable or better privacy with higher testing accuracy, further identifying our approach's effectiveness.

### 5.2.3 Training Cost

The privacy training cost can be optimized via PPTP since PPTP freezes a large portion of the weights. We evaluate the GPU memory and time cost to show the efficiency benefit that PPTP enables through privacy training approaches. As shown in Figure 15, the CUDA memory and time costs, evaluated on NVIDIA Tesla V100, clearly decrease after PPTP is applied. In particular, the actual training time cost is much less than the original training approaches (w/o PPTP) since retraining partial weights requires fewer epochs (less than half epochs) to converge.

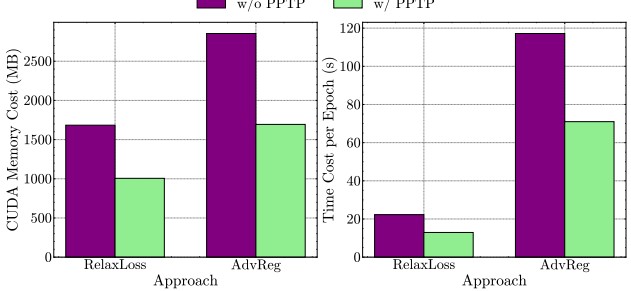

Figure 15: Our proposed Privacy-Preserving Training Principle (PPTP) clearly decreases training costs in memory and time. (CIFAR100, ResNet18).

### 5.2.4 Feature Map Differences

A significant factor that makes the model less generalized with a privacy-defending approach is that the feature representations in each model layer are quite different. As shown in Figure 16, the differences in feature maps between training with RelaxLoss and CE are more significant than training with different data (full vs. half). However, closer feature maps do not mean better generalizability (please refer to Table 7). These identify that our approach can preserve the generalized features learned by CE at privacy-safe layers and help the privacy-risky layers become more privacy-safe to mitigate privacy leakage.

## 6    Conclusion

In this paper, we observed that privacy vulnerability occurs at a portion of layers rather than the entire network. We underscore that the generalizability and privacy risks are decomposable since the well-generalized features and privacy-risky features exist in the different regions of the model. With this insight, we proposed Privacy-Preserving Training Principle (PPTP) to preserve generalizability along with privacy training. Through extensive empirical results, we showed that our approach enhances privacy with the proposed efficient training while not losing generalizability.

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
