# OpenReview forum: "Decoupling Generalizability and Membership Privacy Risks in Neural Networks"
_TMLR — Withdrawn by Authors_

### Review · Reviewer_qrYF · 2025-08-01

**Summary Of Contributions:**

This paper investigates membership inference attacks (MIAs) in vision datasets (TinyImageNet, CIFAR-100) on 3 type of models (VGG, ResNet, ATM-transformers).

With experiments it show that **the later the layer in a neural network, the more privacy risk it hold**. For this, the paper proposes an experiment "Sample-Level Measurement Design", where the training data is split to 2 and 2 models are trained: one with the full dataset, one with only half. The intermediate representation after a given layer is compared between the 2 models on the entire dataset. The more distinct the difference between data points seen by both models and data point seen by only one, the more vulnerable the model architecture is against membership attacks.

Ablation study include adding data augmentation, modifying the feature map size, the channel size and model architecture depth.

On this principle, the paper suggest that privacy-preserving techniques can be applied to only the last layers of the model, improving training time, while keeping competitive or even better privacy-performance trade-off than methods applied on the full model.

Experiments on CIFAR-100 show that the PPTP principle improves the RelaxLoss defense technique against 4 out of 5 attack types, and Adv-Reg on 1 out of 5 attack types, while keeping similar privacy-performance trade-off against the rest. On the TinyImageNet dataset PPTP improves RelaxLoss vs all attack types and Adv-Reg vs 4 out of 5. Combined with SELENA, no significant changes can be ovserved.

**Audience:**

Yes

**Audience Explanation:**

Yes, the paper is relevant for the machine learning community working on privacy and Membership Inference Attacks (MIAs). Recent papers in various venues suggest that this topic is much in interest of the audience of TMLR.

**Claims And Evidence:**

Yes

**Claims Explanation:**

Both hypothesis verification and PPTP retraining principle are demonstrated with extensive experiments on benchmark vision datasets.

**Requested Changes:**

1. On the novelty of **privacy risk occurs in later stages**:
[1] demonstrates that for a convolutional network, the later convolutional layer is used to infer membership the better the attack accuracy is. The literature review fails to identify this core contribution to the paper's research topic.
2. There is a typo on Figure 1 Stage 4 Blcok 3
3. Is there a specific reason why different ResNet versions are used through the paper? (ResNet152 in Figure1, in Section 3.3, ResNet 18&30 in Figure 2, ResNet18 as default for the rest of the experiments). Keeping a consistent experimental setting significantly strengthens the reliability and validity of the analysis.
4. A consistent bin number through Figures should be also considered.
5. In Section 3.3, it states the model contains 33 blocks. This is the block number for ResNet101, not ResNet52
6. An illustration describing the stages of the different model types would help the reader.
7. In multi-layer retraining (5.2.1), the cut-off happens at Stage 4?
8. Were the same experiments done for CIFAR-100 to select the cut-off, or the same layers were used as determined in TinyImageNet? The improvement-difference between Figure 12 and Figure 13 might suggest that it has to be re-calibrated for any datasets.
9. CE is never defined (it is cross-entropy loss, right?)
10. The used MIAs might be outdated (the newest is from 2021), and weak (AUC-ROC curve in Figure 14 is nowhere near LiRA attack's in [2] or trajectoryMIA attack in [3]). It would be nice to see the experiments done with the SoTA methods in [1,2,3]. It would be also interesting to leverage [1] on retrained and not retrained layers.


[1]  Li, Jiacheng, Ninghui Li, and Bruno Ribeiro. "Effective passive membership inference attacks in federated learning against overparameterized models." The Eleventh International Conference on Learning Representations. 2023.

[2] Carlini, Nicholas, et al. "Membership inference attacks from first principles." 2022 IEEE symposium on security and privacy (SP). IEEE, 2022.

[3] Liu, Yiyong, et al. "Membership inference attacks by exploiting loss trajectory." Proceedings of the 2022 ACM SIGSAC Conference on Computer and Communications Security. 2022.

I believe that considering these changes can improve the quality of the paper. I also want to thank the authors for their valuable contribution to science!

---

### Review · Reviewer_2iJ6 · 2025-08-12

**Summary Of Contributions:**

The paper investigated privacy leakage across different layers of a machine learning model. Based on empirical evidence, it claimed that only a small portion of layers (those closer to the output) pose significant privacy concerns, and accordingly proposed a method to enhance privacy.

**Audience:**

Yes

**Audience Explanation:**

Privacy preservation is an active area of research.

**Broader Impact Concerns:**

N.A.

**Claims And Evidence:**

No

**Claims Explanation:**

The empirical evidence contradicts the data processing inequality. There is no way that later layers contain more information than earlier layers (as suggested in the paper, where late layers are claimed to hold private information while earlier layers do not).  The use of the L2 distance to measure disparity is highly questionable. The observed disparity in the later layers is believed to result from the regularization of features in the L2 space, whereas the features in earlier layers are more unstructured, making the L2 distance less meaningful.

**Requested Changes:**

I have no suggestions for changes to improve the paper, as I believe the observations and subsequent analysis are fundamentally at odds with principles of information theory, most likely due to the metric used to measure disparity. I would encourage the authors to assess disparity using metrics such as mutual information, KL divergence, or other information-theoretic measures.

---

### Review · Reviewer_y5Kr · 2025-09-01

**Summary Of Contributions:**

The authors study the difference in latent representations between used and not-used training datapoints in different parts of a neural network and propose a empirical privacy protection that aims at re-training parts of the model that were determined as being leaking privacy using privacy-preserving approaches. The authors attack their solution and other solutions using membership inference attacks and compare the trade-offs.

**Additional Comments:**

- Sec 1: Why are generative models and transfer learning a domain? What do you mean by domain here?
- Sec 2: "models always inevitably have to pay the price of utility for privacy with current approaches", looking at [2] I do not see why this absolute statement would be true.
- Sec.3.1: "A direct advantage of training from scratch is that it is implementation-friendly": I am unsure why transfer learning would be less implementation-friendly or what do the authors mean?
- Fig. 5: Stage is now the same in each row and not in every column which is different from other plots
- Tab. 5: Typo "Pytroch"
- Fig. 11: Add random baseline at 0.5 accuracy, if mia accuracy means that
- Fig. 14: Acc could now mean test accuracy or mia accuracy
- Sec. 5.2.1: "Most privacy defenses [...] are designed for training from scratch. Thus rewinding is more adequate for them." -> Why would resetting to pre-training weights not be feasible? I am not sure if I follow this sentence.

**Audience:**

Yes

**Audience Explanation:**

I think yes, because it is important to understand what parts of machine learning models leak privacy. I believe the TMLR community would be interested in this.

**Broader Impact Concerns:**

In my opinion it is important to mention that this purely empirical privacy protection and that it can only be interpreted as such. This means that it should be used in critical applications with a lot of caution to avoid negative impacts.

**Claims And Evidence:**

No

**Claims Explanation:**

- both "privacy risk" and "generalizability" are not defined
    - what is meant by privacy risk? It would be important to clarify that emperical privacy is meant and not formal privacy and provide an actual definition
    - what is mean by generalizability?
    - Sec 1: "decouples utility and privacy into two separate parts": Could the authors elaborate what is meant by this?

- validation of the method is lacking:
   - SoTA LiRA attack (Carlini et al. (2022a) is not used, despite significantly outperforming the approaches listed in this paper, e.g., see their Fig. 1)
   - no code implementation available, which is critical for privacy defenses as pointed out by prior work (e.g., [0])
   - authors write "For each result, we execute three independent runs to ensure its stability", but figures do not seem to have error bars
   - Sec. 5.2.3: "privacy training cost can be optimized via PPTP since PPTP freezes a large portion of the weights": I do not follow this because it seems according to Alg. 1 that the papers get frozen and then re-trained. Perhaps the authors could elaborate on this and compare to the non-private cost to actually understand the cost

- comparison to SoTA defense method differential privacy is missing:
  - paper only refers to DP-SGD, but does not explain in depth why it is not used
  - DP provides **theoretical** guarantees for privacy and has connection to TPR at FPR for MIA (see Theorem 2.1 of [1])
  - DP can be used to train with high utility [2,3,4]
  - DP prevents success of MIA by a lot (see Fig. 8 of [3] as an example)
  - DP is an example for what you refer to as "non-decorator defense approach" and transfer-learning is used there a lot (see references in [3,4] for a list of these, but note that these are somewhat older works from 2023 and 2024, so there are many newer ones)

- paper is not clearly written:
    - Section 3.2: Unclear why feature map difference is a sensible metric for privacy (or why is it referred to as privacy risk on page 5 under "depth")
    - Section 3.3: Model architecture is unclear (are there only 4 stages or why are subsequent stages not explored?)
   - Section 3.3 ("With no doubt, distributed generalized privacy-safe features...") without reference or evidence for this is a statement without context
   -  Section 5.2: Unclear what MIAs Accuracy is and how it is computed and why it is a sensible metric in comparison to TPR at low FPR (Carlini et al. (2022a))
   - Sec. 5: Explain the other methods, it is not obvious what they are without at least a short explaination

- Other critical points
  - Feldman and Zhang [5]: You might have missed this work, but I think it is critical and very influential work (e.g., their Section 3.6)
  - The paper seems to focus mostly on the average vulnerably and not the worst-case. Is there a motivation for that? How would the extension to the worst-case look like?


--------
  References
 - [0] Ganev, G., Annamalai, M. S. M. S., & De Cristofaro, E. The Elusive Pursuit of Reproducing PATE-GAN: Benchmarking, Auditing, Debugging. In TMLR 2025.
  - [1] Kairouz, P., Oh, S., & Viswanath, P. The composition theorem for differential privacy. In ICML 2015.
  - [2] De, S., Berrada, L., Hayes, J., Smith, S. L., & Balle, B. (2022). Unlocking high-accuracy differentially private image classification through scale. arXiv:2204.13650.
  - [3] Tobaben, M., Shysheya, A., Bronskill, J. F., Paverd, A., Tople, S., Zanella-Beguelin, S., ... & Honkela, A. On the Efficacy of Differentially Private Few-shot Image Classification. In TMLR  2023.
  - [4] Tramèr, F., Kamath, G., & Carlini, N. Position: Considerations for Differentially Private Learning with Large-Scale Public Pretraining. In ICML 2024.
- [5] Feldman, V., & Zhang, C. (2020). What Neural Networks Memorize and Why: Discovering the Long Tail via Influence Estimation. In NeurIPS 2020.

**Requested Changes:**

- define (empirical) privacy risk and generalizability (critical)
- compare to DP (critical, unless there is a reason for not comparing that I do see at the moment)
- use the SoTA attack LiRA (using clearly weaker attack makes one wonder if the defense works)
- provide code (critical for privacy research)
- explain why there are no error bars (strengthen)
- discuss the compared to defenses (critical)
- explain why MIA accuracy is used over TPR at FPR and what it means (critical)
- add and discuss reference to [5] and discuss worst-case [experiment is not needed here, but it should be mentioned] (critical)
- clarify all the other minor points above (strengthen)
- look at the list of typos and questions below under additional points

---

### Note · Authors · 2025-09-03

I have read and agree with the venue's withdrawal policy on behalf of myself and my co-authors.